# A method for non-destructive microwave focusing for deep brain and tissue stimulation

**Vijay Harid** [1]*, **Hoyoung Kim**[2], **Ben-Zheng Li**[1,3], **Tim Lei**[1]

**1** Department of Electrical Engineering, University of Colorado Denver, Denver, CO, United States of America, **2** Space Sciences Laboratory, University of California, Berkeley, Berkeley, CA, United States of America, **3** Department of Physiology and Biophysics, University of Colorado Anschutz Medical Campus, Aurora, CO, United States of America

\* vijay.harid@ucdenver.edu

**Data Availability Statement:** The link to the data containing the FEKO files and lua code used to generated the Green's functions in this study can be found at Zenodo, this is the new DOI: (https://doi.org/10.5281/zenodo.7452128).

## Abstract

Non-invasive stimulation of biological tissue is highly desirable for several biomedical applications. Of specific interest are methods for tumor treatment, endometrial ablation, and neuro-modulation. In traditional neuro-modulation, single- and multi-coil transcranial stimulation techniques in low oscillation frequencies are utilized to non-invasively penetrate the skull and elicit action potentials in cortical neurons. Although these methods have been proven effective, tightly focusing these signals to localized regions is difficult. In recent years, microwave (MW) methods have seen an increase usage as a minimally invasive treatment modality for ablation and neuro-stimulation. Unlike low frequency signals, MW signals can be focused to localized sub-centimeter regions. In this work we demonstrate that a three-dimensional array of MW antennas can be used to tightly focus signals to a localized region in space within the human body with MW frequencies. Assuming an array of small MW loop antennas are placed around the body, the optimal amplitude and phase of each array element can be accurately determined to match an arbitrary desired field profile. The major innovation of the presented method is that the fields that penetrate the biological region are determined via computing numerical Green's functions (NGF) that are then used to drive an optimization algorithm. Using simplified models of regions in the human body, it is shown that the MW fields at 1 GHz can be focused to sub-centimeter sized "hot spots" at depths of several centimeters. The algorithm can be easily extended to more realistic models of the human body or for non-biological applications.

## Introduction

Electrical stimulation is widely used in biomedical applications ranging from subcutaneous therapeutic treatments for pain relief to stimulating brain nuclei for improving motor symptoms of Parkinson's disease patients [1, 2]. These electrical stimulation techniques modulate neural activities through direct current injection to neurons, and therefore some of these methods require invasive craniotomies to implant electrodes near the intended neurons for stimulation [3, 4]. For instance, Deep Brain Stimulation (DBS) requires insertion of an extended electrode deep into the mid-brain to stimulate the sub-thalamic nucleus, which can

**Funding:** The author(s) received no specific funding for this work.

**Competing interests:** The authors have declared that no competing interests exist.

result in various hardware complications [5]. As such, non-invasive neural modulation techniques are extremely useful for expanding the selection of neural stimulation technologies.

In general, the various non-invasive stimulation methods can be classified into (i) low frequency stimulation, and (ii) high frequency (microwave) stimulation. Each of the two classes require distinct engineering design procedures and often have different applications [6, 7]. To adequately motivate the findings in this work, a brief discussion of the relevant properties of low and high frequency systems are presented. Low frequency methods typically utilize radio signals in the frequency band of 0–3 kHz [8]. For instance, Transcranial Magnetic Stimulation (TMS) is an FDA approved treatment modality in which an unfocused magnetic field is transmitted through the brain to alleviate symptoms of several neurological disorders, including depression, Alzheimer's, and Parkinson's disease [9, 10]. Low frequency signals are particularly useful because of their ability to penetrate the skin, tissue, and bone [11, 12]. However, spatially focusing with low frequency waves can be difficult and requires large and complex coil designs [13, 14].

There have been recent efforts to employ multi-coil configurations to interfere low-frequency magnetic fields emitting from the coils to localize low-frequency magnetic fields for neural stimulation within the brain. In the work by Jiang et al. [15], a two-dimensional wire mesh was numerically simulated to evaluate magnetic field confinement on the cerebral cortex. By controlling the currents flowing through the wires of the two-dimensional mesh, a local loop current can be formed on one of the grids of the two-dimensional mesh to induce a local magnetic field on top of a small portion of the cerebral cortex. Recently, Navarro de Lara *et al.* [16] designed and evaluated a 3-axis coil through interlocking three magnetic coils perpendicular to one another along the three geometrical axes. The 3-axis coils were then used to cover a head surface to simulate magnetic field localization within the head. Their simulated results indicated that magnetic field energy can be localized onto one of the cortical sulci. However, the low frequency of the magnetic field limits the focusing capability to a broad region on the cerebral cortex. Magnetic focusing to deeper brain regions, such as nuclei in the mid-brain, were not demonstrated. Osanai *et al.* [17] used submillimeter-sized coils to generate a weak magnetic field for neural stimulation. Their work demonstrated that a single millimeter magnetic coil can generate strong enough magnetic field to excite mouse auditory cortex neurons reaching down to layer 6. In addition, a multiple coil version was developed and incorporated into an invasive neural probe. Through magnetic interference, magnetic field on top of the probe can be moved locally through adjusting the currents of the coils. In summary, all these multi-coil approaches are largely optimized for lower magnetic field frequencies; thus, physically limits the ability for tight magnetic focusing deep within the brain.

High frequency methods for tissue stimulation typically utilize radio signals in the microwave band (100 MHz– 10 GHz). The most commonly used high frequency system is microwave (MW) ablation. MW ablation typically relies on the surgical insertion of a MW antenna which consequently stimulates tissue via conduction heating. Although traditional MW ablation typically does require surgery, non-invasive versions of the system have been tested for select cancer treatment applications [18]. Although still poorly understood, MW radiation has shown to impact neural activity in various studies [19]. For instance, [7] showed that modulated MW signals can also be used to vary firing rates of neurons in mice; however, the phenomenon was demonstrated under invasive laboratory conditions. Thus, MW stimulation systems can serve as a potential alternative to the more predominant low frequency methods.

In contrast to low frequency methods, MW signals have the distinct advantage of a high degree of spatial focusing. The shorter wavelengths of MW signals allow for engineered interference patterns or detection of sub-centimeter sized tumors [20]. For instance [5], utilized a MW array and space-time beamforming for the detection of breast cancer tumors less than 2

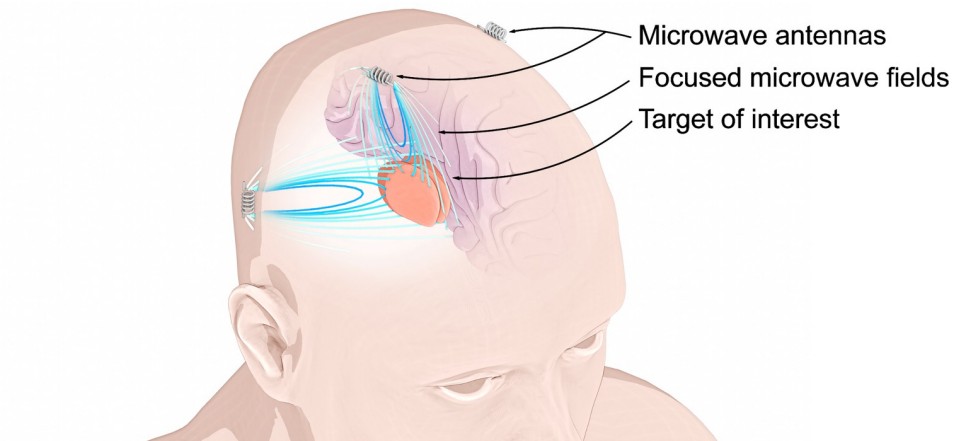

**Fig 1. Illustration of non-invasive MW focusing based on magnetic inference with small coil (magnetic dipole) antennas placed near the surface of the head.**

mm in size. However, three-dimensional MW focusing within biological tissue has not been adequately investigated in the past due to the complexity of modeling MW signal propagation through the human body. MW signals, once inside the human body, will have wavelengths that are shorter than in free space and can exhibit complicated scattering and diffraction. Simultaneous focusing and stimulation of desired regions in the human body opens the door to a localized and non-invasive stimulation modality in a manner that has not been possible in the past. In this work, a method for focusing MW signals within biological tissue is presented by using advanced numerical electromagnetics techniques. Using electromagnetic simulations, we demonstrate that a three-dimensional array of MW coil antennas can be used to focus signals to sub-centimeter regions within the human body, as illustrated in Fig 1. Section II describes the relevant physics behind microwave propagation in tissue along with the numerical methodology employed for computing MW fields within a volume of inhomogeneous biological material. Section III describes an optimization method that determines the required electrical current in each MW antenna such that the signals can be focused via three-dimensional constructive and destructive interference. Section IV shows examples of focused MW fields in a simplified brain and limb model; the ability to electronically steer the focused MW field is also demonstrated. Section V discusses potential improvement to the system and describes biomedical applications that are relevant to the presented results.

## MW penetration and focusing in biological materials

In order to accurately understand microwave penetration and focusing in biological tissue, it is necessary to first introduce the relevant mathematics that describes the physical system. The following two subsections describes the physics of microwave signals in lossy media and the interference patterns created by multiple MW transmitters.

### Microwave penetration in biological tissue

In order to accurately model the penetration and focusing of MW signals inside biological tissues, it is useful to discuss the relevant parameters and mathematical formalism that governs the physics of the problem. Specifically, the amount of signal penetration is entirely determined by the electrical conductivity $\sigma$, dielectric permittivity $\epsilon$, and MW frequency $f$ [21, 22]. Typical values of conductivity and permittivity in the MW frequency band are shown in

**Table 1.**

| Material | Property | 100 MHz | 1 GHz | 10 GHz |
|---|---|---|---|---|
| **Brain** | $\epsilon_r$ | 89.8 | 48.9 | 34.6 |
| | $\sigma(S/m)$ | 0.79 | 1.31 | 9.78 |
| **Skull** | $\epsilon_r$ | 27.6 | 20.6 | 12.7 |
| | $\sigma(S/m)$ | 0.173 | 0.364 | 3.86 |
| **Muscle** | $\epsilon_r$ | 66 | 54.8 | 42.8 |
| | $\sigma(S/m)$ | 0.708 | 0.978 | 10.6 |
| **Blood** | $\epsilon_r$ | 76.8 | 61.1 | 45.1 |
| | $\sigma(S/m)$ | 1.23 | 1.58 | 13.1 |
| **Fat** | $\epsilon_r$ | 12.7 | 11.3 | 8.8 |
| | $\sigma(S/m)$ | 0.0684 | 0.116 | 1.71 |

Table 1 [23]. With these aforementioned parameters, the effective penetration depth or "skin depth", $\delta$, is given by,

$$\delta = \frac{1}{\sqrt{2\pi\mu\sigma f}} \tag{1}$$

Here, $\mu$ is the permeability of free space. In addition, the wavelength, $\lambda$, inside the biological material is given by,

$$\lambda = \frac{1}{f\sqrt{\mu\epsilon}} \tag{2}$$

In general, the skin depth and wavelength both decrease as either conductivity or frequency is increased. Higher frequencies allow for short wavelengths that strongly aid in focusing signals to a small spot size. However, the high frequency signals simultaneously undergo rapid attenuation due to the corresponding short skin depth. This basic concept defines the physically limiting tradeoff for MW focusing in biological tissue.

Fig 2 shows skin depth and wavelength as a function of frequency for material properties that correspond to several different biological materials [24]. As shown, in the range of 100 MHz to 10 GHz, MW signals can penetrate between 10 cm and 30 cm depending on the frequency and material. The ranges of skin depth values in this frequency band are quite reasonable given the typical dimensions of the human body and serve as a starting point for designing the transmit signals. However, even with selecting the appropriate frequencies for penetrating deep into tissue, focusing the signals to a spatially-limited spot size remains a challenging task.

## Method for calculating microwave fields from multiple transmitters

The typical method of focusing MW signals is by utilizing an antenna array [25, 26] and is also the methodology employed in this work. In most array applications, however, the MW is in the form of a *beam* (or multiple beams) akin to a laser [27, 28]. Thus, even though the signal may be just a few millimeters in spot size, the entire path from transmitter to the desired region of stimulation is illuminated. For deep tissue stimulation applications, however, illuminating the entire "line-of-sight" will impact non-target areas and is undesirable. For such scenarios, novel methods for array and signal design are needed to create a MW "hot-spot" that only illuminates a specific three-dimensional region while minimally effecting the surrounding area.

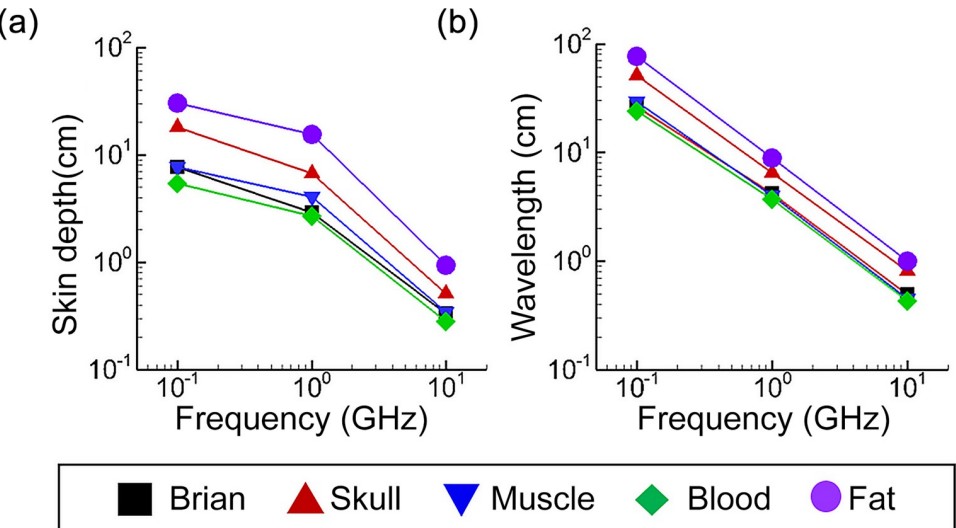

**Fig 2. Skin depth and wavelength as a function of frequency for brain, skull, muscle, blood, and fat properties.**

Optimizing a MW array for focusing inside an arbitrary geometry is a difficult engineering task. One approach to do so is via utilizing a Green's function formalism. In the context of electromagnetic theory, the Green's function represents the radiation (either electric or magnetic fields) at an arbitrary position due to an infinitesimal source. In general, an infinitesimal source be thought as a combination of point electric or magnetic dipoles. For the sake of simplicity in this study, it is assumed that all transmitters are small loop antennas (as often used in neural stimulation systems) and can thus be well approximate by an infinitesimal magnetic dipole with moment $m_n$. In particular, loop antennas have a low near-field impedance and as such allow for better transmission through conductive media such as skin and tissue [29]. It is worth noting that although point magnetic dipoles are considered in this study, an arbitrary antenna can also be utilized with its own corresponding Green's function. In empty space, the Green's functions can be computed analytically in a straightforward manner [30]. Within complex materials and geometries, such as biological tissues with different layers, the Green's functions do not have closed form expressions.

In the general case of antennas radiating in the vicinity of any complex material, the Green's functions need to be calculated using an electromagnetic simulation program. An object with the desired material properties and geometry can first be built using CAD software. Then, a discrete spatial 3D grid can be constructed such that the object is completely contained within the grid. For a fixed radiating antenna, the electric and magnetic fields can be computed everywhere on the grid, which is effectively same as calculating the Green's functions. In this work, the Green's functions are computed numerically using the commercial electromagnetic simulation package, FEKO. FEKO utilizes a hybrid finite element and boundary element methodology to compute electric and magnetic fields in arbitrary material structures due to arbitrary sources. The Green's functions are calculated for $N$ sources (for each geometry considered) and stored in a look-up table.

Once the Green's functions are pre-computed, the magnetic (and/or electric) fields for any general antenna configuration can be easily computed. The magnetic field $H_n(r)$ at the location $r$ due to a transmitter located at $r_n$ is given by $H_n(r) = \hat{G}(r, r_n)m_n$. The quantity $\hat{G}(r, r_n)$ is known as the dyadic magnetic Green's function and represents the mapping from the transmitting dipole to the observed magnetic field at position $r$. For a collection of $N$ transmitters,

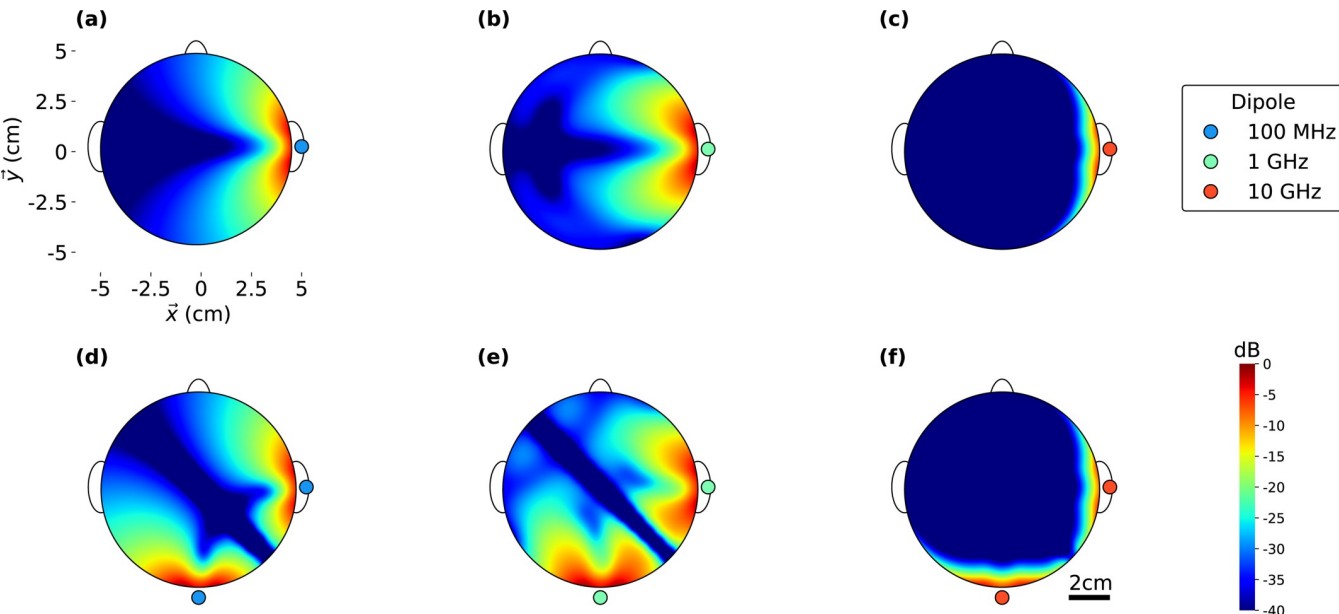

**Fig 3. Magnetic field spatial distributions of a single (a to c) and two magnetic (d to f) dipoles placed outside of a simulated brain-like sphere at 100 MHz, 1 GHz and 10 GHz.**

the total magnetic field can then be computed via a linear superposition of the fields of all the dipoles. Thus, the total magnetic field can be generalized to,

$$\boldsymbol{H}(\mathbf{r}) = \sum_{n=1}^{N} \hat{G}(\boldsymbol{r}, \boldsymbol{r_n}) \boldsymbol{m_n} \tag{3}$$

To illustrate the utility of computing numerical Green's functions using FEKO, the problem of a homogenous sphere with a 5 cm radius is first considered. The sphere is composed of material that corresponds to typical values of brain material as shown in Fig 1.

Fig 3 shows the magnetic field calculated using FEKO simulations for 100 MHz, 1 GHz, and 10 GHz transmit frequencies. At these frequencies, the skin depths are 8.3 cm, 4.1 cm and 0.4 cm respectively as shown in Fig 1. Fig 3(A)–3(C) shows the magnetic field magnitude for the three frequencies and a single $-x$ oriented dipole located 1 cm offset from the surface of the sphere at $x = 6$ cm. Thus, the field profiles are visualizations of the Green's function for the case a homogenous "brain-like" sphere. As shown, 100 MHz, and 1 GHz cases show several centimeters of penetration into the sphere while the 10 GHz case is very quickly shielded. This feature is expected based on the calculated skin depth, suggesting that the lower two frequencies are most useful for focusing deeply inside the material. In order to illustrate the ability to produce arbitrary three-dimensional profiles, Fig 3(D)–3(F) show the electric field in the presence of two magnetic dipoles. The first dipole is the same as before and is located at $x = 6$ cm, $y = z = 0$ cm. The second dipole is oriented in the $-y$ direction and is located at $x = z = 0$ cm, $y = 6$ cm. As shown for the 100 MHz and 1 GHz cases, a 3D field-pattern is created within the sphere with distinct regions of constructive and destructive interference between the fields of the two transmitters. The hot spots are smaller and more frequency at 1 GHz due to the shorter wavelength associated with the higher frequency. This concept is the starting point for engineering a focused interference pattern to create a desired MW *hot spot* in a specified region of interest. The following section describes the optimization algorithm that determines the appropriate transmitter strengths to match a prescribed hot spot profile.

## Method for MW focusing using an optimized 3D antenna array

3D MW focusing of a desired electric or magnetic field profile can be achieved through optimizing the magnitudes and orientations of multiple transmitting dipoles $\boldsymbol{m}_n$ surrounding the biological tissue. The electric field is preferentially chosen for analysis due to its close relationship to conduction currents and consequently neural stimulation. Since the mapping between the transmitters and electric field is linear based on Green's functions, the optimal electric field can be written as,

$$\boldsymbol{E}_{mod} = \boldsymbol{G}_E \boldsymbol{M} \tag{4}$$

Where $\boldsymbol{E}_{mod}$ is a $(1 \times 3N_d)$ stacked vector containing modeled the electric field values (all three components) at $N_d$ grid points within the target of interest. The quantity $\boldsymbol{M}$ is a $(1 \times 3N_t)$ stacked vector containing all $N_t$ magnetic dipoles. $\boldsymbol{G}_E$ is matrix of size $(3N_d \times 3N_t)$ which contains all the Green's functions mapping the $N_t$ transmitter values to the electric field to the $N_d$ grid points. There are many potential methods that can be employed to determine the optimal amplitudes of each of the magnetic dipoles. The most common approach is to formally define an "error" between the model and desired field profile, then numerically determine the magnetic dipole amplitudes that minimize the error. For this study, the optimization is formalized through minimizing a regularized least square (RLS) error function with Tikhonov regularization [8, 21]. Specifically, the error function to minimize is given by,

$$\Delta \boldsymbol{E} = \left\| \boldsymbol{E}_{mod} - \boldsymbol{E}_d \right\|^2 + \alpha \left\| \boldsymbol{M} \right\|^2 \tag{5}$$

where $\alpha$ is the regularization parameter. The regularization term is to ensure numerical stability, and when $\alpha$ is set to zero, the equation above reduces to a simple least square difference between the optimal and desired electric fields. The regularization also ensures a unique solution for cases where the problem would otherwise be ill-posed. With this formalism, the optimal configuration of magnetic dipoles, $\boldsymbol{M}_{opt}$, is given by [31],

$$\boldsymbol{M}_{opt} = \left( \boldsymbol{G}_E^T \boldsymbol{G}_E + \alpha \boldsymbol{I} \right)^{-1} \boldsymbol{G}_E \boldsymbol{E}_d \tag{6}$$

Where $\boldsymbol{I}$ is the identity matrix and the superscript '$T$' corresponds to the transpose operator. In this manner, the Green's functions are computed numerically while optimum electric field is calculated using the closed form expression shown in (6).

The optimization strategy for a specific case is illustrated by Fig 4 in which 64 magnetic dipoles surround a homogeneous sphere of a 5 cm radius having the same electric properties of the human brain. Point magnetic dipole transmitters (or small coil antennas) are represented by small red spheres positioned 1 cm away from the sphere surface are uniformly distributed around the brain-sphere. Specifically, each dipole transmitter is located at position $r_n$ with dipole moment $\boldsymbol{m}_n$ The optimization routine adjusts the magnitudes and the pointing directions of the magnetic dipoles in order to match to the desired electric field profile and achieve a focused electric field profile. Fig 5 shows the desired and optimal electric fields for the homogenous brain-like-sphere. The frequency of transmission is set to 1 GHz for the simulation. In this example, the desired electric field $\boldsymbol{E}_d$ corresponds to a z-directed 1 cm-sized hotspot in the center of the sphere at $x = y = z = 0$. Using the optimal solution in (6), a tightly focused electric field $\boldsymbol{E}$ could be formed in the sphere center with a field profile that is very close to the desired field distribution.

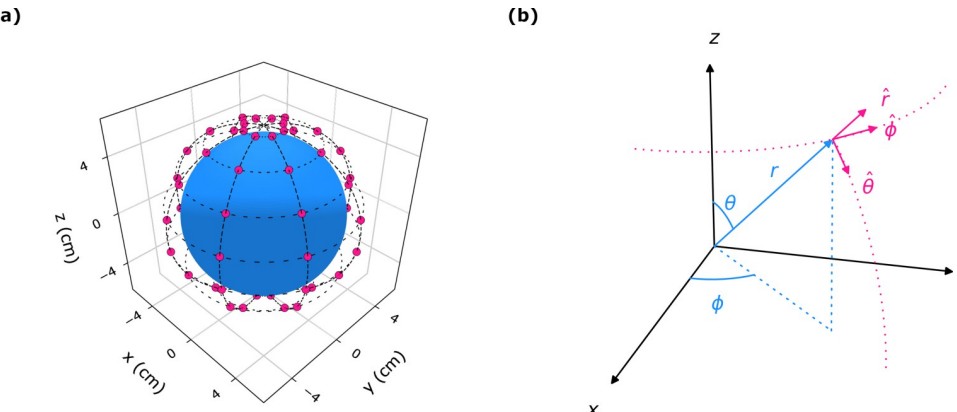

**Fig 4. (a) Multiple magnetic dipoles located at 1 cm away from the brain material.** The shape in the array of 64 dipoles is a sphere with radius 6 cm around a brain of 5 cm radius. Red circles and blue sphere represent the dipoles and a brain material.

## Results

Using the multiple magnetic dipole transmitters and the RLS optimization paradigm, the solver was setup to determine dipole configurations for microwave focusing within various tissue geometries. In these evaluations, it is determined that not only tight electromagnetic focusing inside a biological tissue is possible, the focal point can be electronically steered by changing the dipole configuration. It is worth mentioning that in all cases, the peak desired electric field is set to 0.01 V/m. As described in the previous section, the mathematical description of microwave penetration into tissue is assumed to be linear (i.e heating effects are negligible). As such, if the peak value needs to be higher in an application, all the antenna amplitudes can simply be scaled proportionately to obtain the new peak value. The following three subsections describe the results for a simulated brain scenario and a simulated limb scenario using varying numbers of transmitters and different desired field profiles.

### Microwave focusing in a simulated brain

Fig 6 illustrates the results of microwave focusing at the center of a simulated spherical brain with different numbers of magnetic dipole transmitters. The simulated spherical brain is constructed with three different concentric layers of biological tissues—a brain (inside), a shell-

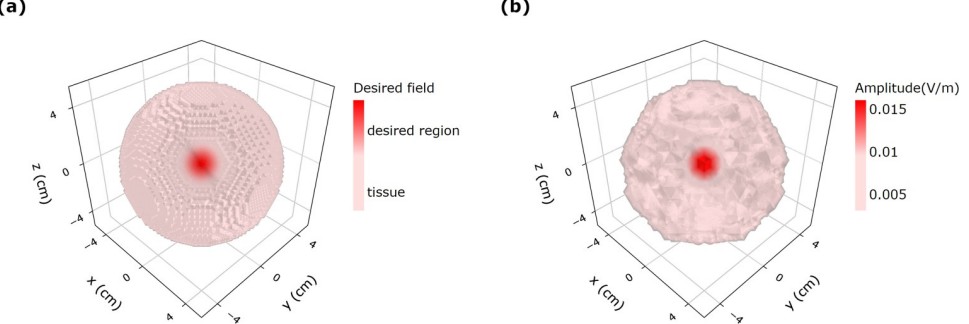

**Fig 5. (a) Desired and (b) optimal electric fields with 64 magnetic dipoles at 1 GHz for brain material.** The radius of the sphere 5 cm and the target point is x = y = z = 0 with a 1 cm spherical spot size.

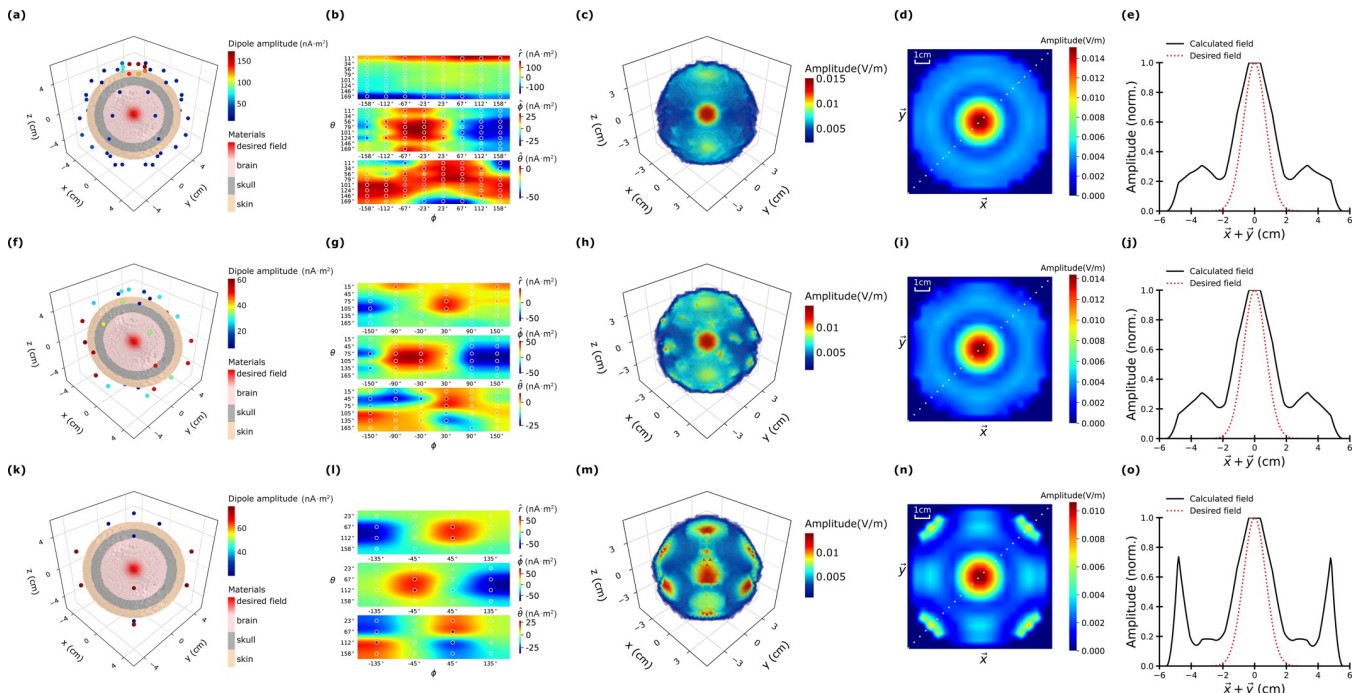

**Fig 6. Results of MW focusing simulation in a 5 cm radius simplified brain model.** The brain model consists of a brain-matter region with radius 3.2 cm at the center of a membranous skull region with 1 cm thickness, and a membranous skin region with 0.8 cm thickness. (a, f, k) show all the dipole locations and amplitudes (shown by color) superimposed on the brain model. The desired field intensity is also shown as the red region at the center of the brain. (b, g, l) are the radial ($\hat{r}$), declination ($\hat{\theta}$), and azimuthal ($\hat{\emptyset}$) components of the dipole amplitudes as function of the angles around the head ($\theta$ and $\phi$). (c, h, m) and (d, i, n) are the electric field amplitude in 3D and a 2D cross-section. (e, j, o) is the field amplitude of optimized field and desired field as a function of the coordinate $\frac{(\vec{x}+\vec{y})}{\sqrt{2}}$. The coordinate $\frac{(\vec{x}+\vec{y})}{\sqrt{2}}$ is along a diagonal direction shown by the white dotted line in (d, i, n). The three rows (counting from top to bottom) correspond to the case of 64, 32, and 16 transmitting dipoles respectively.

like skull (middle), and a skin (outside) layer with the material properties outlined in Fig 2. The radius of the brain is 3.2 cm at the center and membranous skull region with 1cm thickness and a membranous "skin" region with 0.8 cm thickness. It is worth noting that the "skin" layer has made thicker than reality to allow for interspersed adipose tissue. This makes the simulation results even more conservative since skin conductivity is typically higher than that of fat. The simulations considered three different numbers of transmitters corresponding to 64 (panels a-e), 36 (panels f-j) and 16 (k-o) magnetic dipoles. The emitting frequency was set to 1 GHz in the simulation and the desired fields was set to be a 3D Gaussian hotspot with standard deviation of 1-cm located at the center of the simulated spherical brain at $x = y = z = 0$.

The left-most column of Fig 6 (panels a, f, and k) shows the geometrical arrangements of the magnetic dipoles false-colored with the strength of the magnetic dipoles. The second column (b, g, and l) shows the magnitudes of each component ($\hat{r}, \hat{\theta}, \hat{\phi}$) of the magnetic dipoles in an unwrapped 2D representation. It is interesting to note that the only a concentrated clusters of magnetic dipoles had high amplitudes, indicating that much fewer magnetic dipoles may be used in real physical implementation, provided that the location of the desired focus does not need to be changed.

Fig 6A–6E shows that 64 transmitters are surrounded to focus the field at the center in brain model. Fig 6A, 6C and 6E compare the optimal electrical field profiles in 3D, a 2D-slice and a 1D-slice respectively. The dipole amplitude on the top and bottom of the brain is 2 times higher than other location as shown in Fig 6A. Fig 6B shows the radius ($\hat{r}$), theta ($\hat{\theta}$), and ($\hat{\emptyset}$)

components of magnetic amplitude as a function of theta and phi. Among 3 components, the radius component is dominant and has high amplitude at $\theta = 11°$ and $\theta = 169°$ which are the locations on the top and bottom of the brain material. The amplitude of the phi components have the highest positive and negative values at -30 and 150 degree symmetrically. The theta amplitude has broadly high values as a function of phi direction. The dipole amplitudes induce the focal field area as shown in Fig 6C and 6D. Fig 6E shows the field amplitude as a function of $\frac{\vec{x} + \vec{y}}{\sqrt{2}}$. The $(\vec{x} + \vec{y})$ direction represents a diagonal direction as shown by the white dotted line in Fig 6D. The corresponding field in that direction (along with the desired field) is shown in Fig 6(E). At the center of the domain, the amplitude has the highest values and is approximately 5 times higher than 2 cm away from the center. Although not shown, the phase shift between neighboring antennas (for a given polarization) is on the order of 0.2–2 radians with larger phase shifts for the simulations with fewer antennas. The exact phase values emerge as a result of the optimization from (6).

The microwave focus generated by the 64-dipole configuration (panels c-e) best match the desired field with the field strength in the fringe regions (outside desired spot size). As the number of transmitters are decreased from 64 to 32 (Fig 6F–6J), the fields are still similar to that of the 64-dipole configuration, however, additional sidelobes are created due to the reduced number of transmitters. In contrast, the 16-dipole configuration could still focus the microwave fields at the center, but unlike the 64 and 32 dipole configurations, it had several additional sidelobes at outside the desired target hotspot (as shown in panel m). The results show that a larger number of dipoles help cancel out undesired fields via optimized interference. The actual number of transmitters used in practice may be limited by cost and size; however, the simulation results suggest that arbitrary field profiles may be generated with a sufficient number of controlled transmitters. It is important to note, however, that a more complex brain geometry may still require a larger number of transmitters to counteract the higher degree of inhomogeneity.

The 1D intensity cut plots on the far right of Fig 6 demonstrate how tightly the microwave fields can be focused. Specifically, the FWHM (Full-Width-at-Half-Maximum) for the 64, 32, and 16 dipole transmitter configurations were measured to be 2, 2.2, 3.4 cm. The values for the 64 and 36 dipole cases are relatively close to the true 2.36 cm FWHM value of the desired Gaussian hotspot profile.

The simulation results thus demonstrate a proof-of-concept for non-invasive neural stimulation using an array of optimized microwave transmitters. Although a simplified brain model was employed in these simulations, a more sophisticated brain model can be easily included within the same modelling framework. The following subsection shows another set of simulated results in the case of a limb-like geometry.

## Microwave focusing in a simulated limb

Microwave focusing can also be employed for cylindrical shaped limb-like structures. Fig 7 shows a simulated cylindrical leg model with a radius of 5 cm and a length of 20 cm. Similar to the simulated brain, the leg model consisted of a bone, a muscle, and a tissue layer in concentric cylindrical shapes. The bone radius is 1 cm bone radius, the muscle is muscle 2 cm thickness, and "tissue" is 2 cm thickness. Once again, the tissue layer has made thicker to approximate both skin and interspersed adipose tissue. 64, 36, and 16 magnetic dipoles uniformly surrounding the limb model were simulated to evaluate 1 GHz microwave focusing on one side of the limb model. The focused spot is 2 cm away from the center point which mimics subdermal nerve stimulation in clinical scenarios such as pain relief.

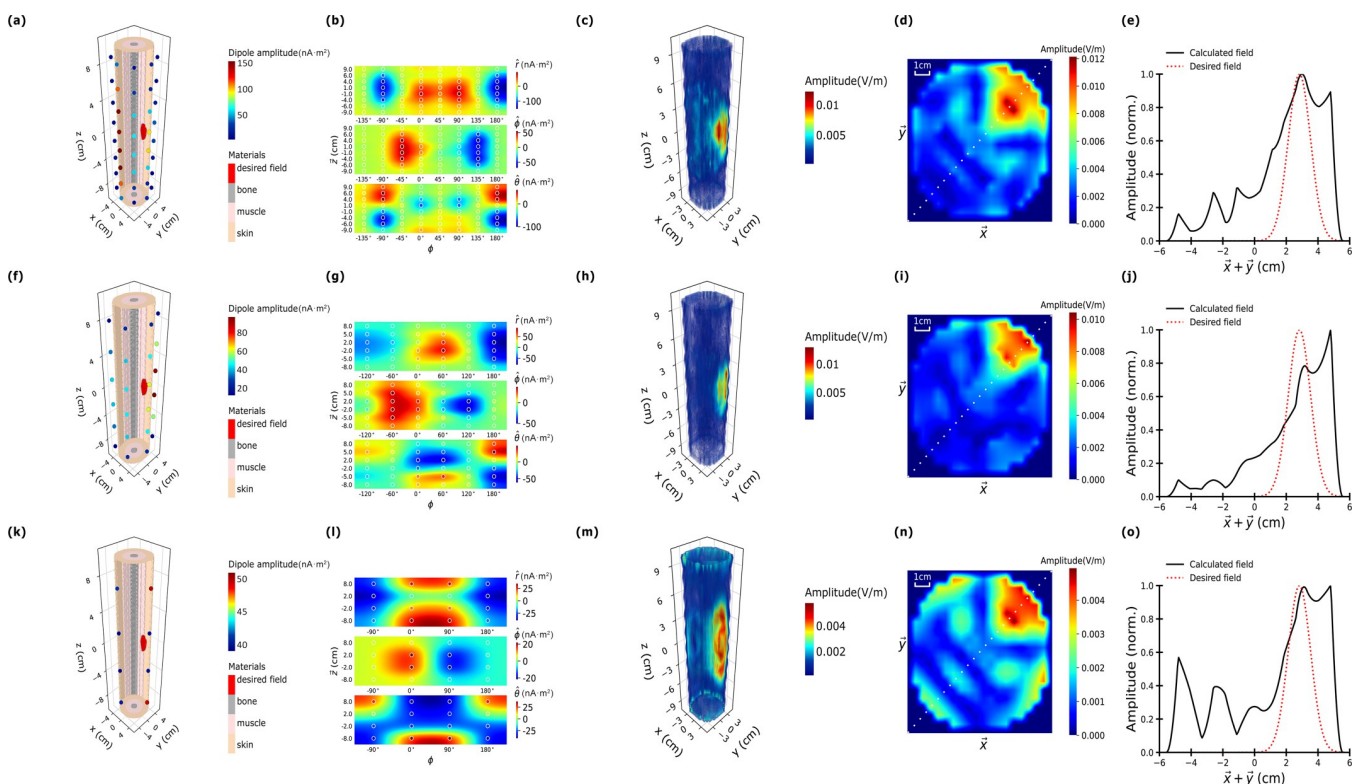

**Fig 7. Cylindrical leg model with radius 5 cm and length 20 cm using (a), (d), (g) 64 dipoles, (b), (e), (h) 32 dipoles, and (c), (f), (i) 16 magnetic dipoles at 1 GHz.** The leg model consists of cylindrical bone region 1 cm radius at the center, membranous muscle region 2 cm thickness, and membranous tissue 2 cm thickness. The format is the same as that of Fig 6.

The leftmost column of Fig 7 (panels a, f, and k) illustrates the geometrical arrangements of the 64, 36 and 16 magnetic dipole emitters with the optimized emitting dipole strengths are false-colored for visualization. As shown the dipoles closest to the desired focal point have the highest intensities. The second column (panels b, g, and l) illustrates the field intensities and the orientations of the dipoles in an unwrapped 2D representation. This 2D view clearly shows that dipoles over a large range of positions have moderate to high intensities and are not limited to only regions directly around the focal point. This is a natural outcome of the optimization procedure and is vital to eliminate side-lobes through destructive interference. Even so, less than half of the transmitters have appreciable intensities, suggesting that either fewer transmitters could be used in principle.

The three columns on the right illustrates the optimal field distribution in 3D, 2D intensity and 1D cut plots. Similar to the spherical brain, well-focused microwaves can be achieved for the 64 and 36 dipole configurations. The 16 dipole case could not generate the same contrast ratio as in the other two cases. This is likely because the geometry of the problem makes it mathematically difficult to cancel out sidelobes and create a highly peaked signal simultaneously with only 16 dipoles. As shown in Fig 7(K), only 4 of the 16 dipoles have appreciable amplitudes, which makes it difficult to achieve the desired field profile.

The results show that microwave fields can be focused inside a cylindrical geometry, however, a high density of transmitters is required to ensure adequate focusing.

## Non-mechanical scanning of the focal point

Fig 8 demonstrates that focused microwave fields can be swept through space by changing the intensity pattern of the dipole array. In this scenario, 64 magnetic dipole antennas were used uniformly distributed across the simulated brain sphere with the same geometry as Fig 5. Three focal targets—x = y = z = −1 cm (lower left), x = y = z = 0 cm (center) and x = y = z = 1 cm (upper right)–are presented to the optimization routine individually to seek the required dipole intensities. The optimization routine successfully found all three transmitting patterns with tight focal points at all three desired target locations. However, the two off-centered magnetic foci had a 50% intensity reduction compared to that of the center. The FWHM for the center focus was ~2 cm and the off-centered foci were slightly asymmetric with a FWHM of ~3 cm. Overall, the results suggest that by electronically varying the transmitting antenna intensities, a microwave hotspot can be swept spatially across a desired target area.

## Discussion and future work

The results presented in this work show that a microwave hotspot can be theoretically generated inside a target of interest by appropriately driving an array of antennas that surround the region. The methodology sets the stage for non-invasive stimulation of biological tissue and is thus a critical step for several applications. Even so, the current method serves primarily as a general framework and several important points of improvement are worth discussing.

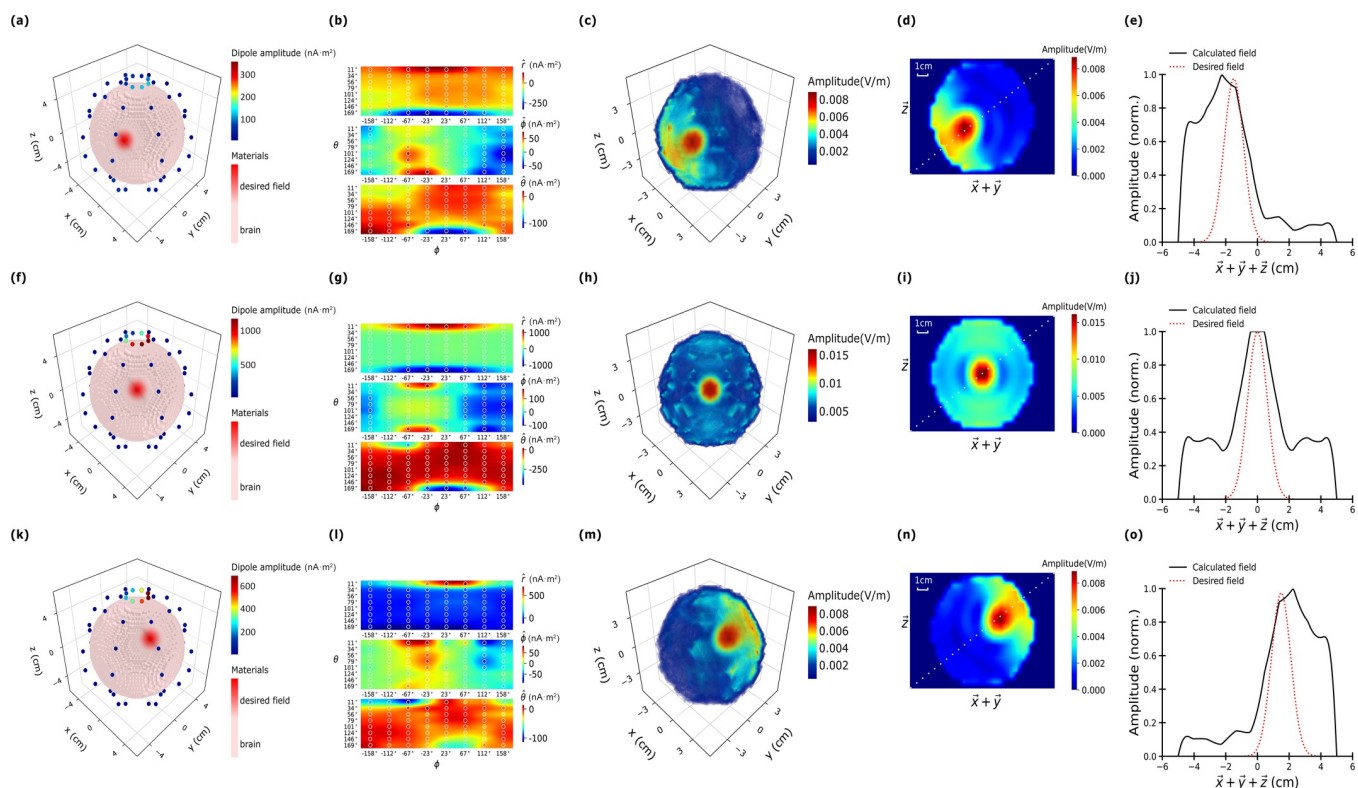

**Fig 8.** Spherical brain model with only brain structure with radius 5 cm using 64 dipoles at 1 GHz for different target points at (a), (d), (g) x = y = z = −1 cm, (b), (e), (h) at x = y = z = 0, and (c), (f), (i) x = y = z = 1 cm. The material properties are consistent with gray matter and the amplitude of focusing MW field is normalized.

In the current methodology, the locations of transmitting dipoles are fixed independently of the desired field profile. The reason for this choice is that fixed transmitter locations make the optimization problem linear. If the dipole locations are treated as unknown, the optimization problem becomes nonlinear and requires more sophisticated solution techniques. However, as seen in all the simulations shown in Section IV, only a fraction of the available dipoles are at appreciable levels of intensity. As demonstrated in Fig 5B where MW signal was focused inside a spherical brain, only magnetic dipoles concentrated on the top and the bottom of the brain sphere have high intensities while the intensity of dipoles around the sphere are greatly reduced. Based on these results, MW focusing can be realized with far fewer magnetic dipoles in practice, especially when the magnetic focus is stationary. The optimization outcomes could thus be greatly improved by employing a nonlinear formalism or including an iterative framework.

In the current framework, each transmitting antenna is assumed to have a magnitude and phase that is independently controllable. Practically, driving each antenna with its own input can be a difficult engineering task. However, in the past several years, advances in antenna array technology makes the possibility of controlling hundreds of antennas quite viable. For instance, active electronically steered arrays (AESA) are routinely used in modern radar and MIMO systems. As such, aside cost limitations, the optimization methodology described herein is reasonable to implement with existing array technology.

All the simulation results shown have only considered a frequency of 1 GHz due to appreciable levels of penetration. This choice has shown to be a viable option, however, different regions of the body might be better suited for higher or lower frequencies. As such, future optimization solutions may also consider keeping the frequency as an unknown parameter that needs to be found. This could further improve the optimization results and provide a higher degree of balance between focusing and penetration. Furthermore, time-domain methods [28, 32] may more easily allow for nonlinearities or heating due to potentially large amplitude fields. Several alternative options of electromagnetic solution methods can be considered in order to leverage desired attributes.

An important point is that the spherical and cylindrical geometries shown in this work do not limit the applicability of the optimization method. The advantage of computing Green's functions numerically allows arbitrary geometries and distributions of material properties. The problem is confounded by the fact that the geometry and material properties may not be entirely known a-priori. In such scenarios, the microwave focusing methodology should be supplemented other imaging modalities [33] to first obtain the required properties of the biological system. Future work may also consider extending the microwave optimization methodology to an imaging system.

Controlled MW focusing has far-reaching implications for clinical applications of neural stimulation. For instance, surgeries of deep brain stimulation are currently only reserved for patients who have developed advanced drug-resistive symptoms since the procedure is otherwise considered excessively invasive. Therefore, developments of next generation deep brain stimulation techniques, especially in a non-invasive manner, may allow the extension of this therapeutic approach to less severe patients or to control disease (such as Parkinson's) at a much earlier Stage. Furthermore, current non-invasive neural stimulation methodologies, including multi-channel TMS, can be difficult to focus a magnetic signal tightly to a desired location in the brain due to the low frequency nature of the signals. However, MW focusing in combination with low frequency amplitude modulation could permit both sub-centimeter scale focusing and neural stimulation ability simultaneously. Additionally, as demonstrated in Fig 7, non-invasive MW focusing in multiple locations may be possible and can open up new opportunities to study neuronal circuits with more complex neuronal modulations and potentially better neurological disease control.

Besides MW focusing, interfering low frequency electric fields [34] and focused ultrasound [35] are two new modalities that have successfully been demonstrated to locally modulate neural activities in deep brain areas. Comparing between the three techniques, the dielectric constants of brain tissues at high MW frequencies are largely uniform and requires no tissue contacts which may simplify the complexity of focusing and less immune to structure fluctuations between different human subjects. While low frequency electric fields are a more direct stimulation mechanism for modulating neuron firing activities, the low frequency nature of the technique requires good probe contacts onto the scalp to drive currents and may be inconvenient for certain applications. Unlike electric and magnetic field stimulations which largely modulate charged ions, ultrasound creates vibrational waves of tissue and fluid molecules of neurons [36], which may cause local heating if high power ultrasound is used. In addition, ultrasound is highly absorptive to bone or skull structures and relatively high power (tens of watts) of ultrasound are required [37]. On the other hand, the maximum permissible exposure limits for electric field can be as high as hundreds of V/m and for magnetic field is up to several A/m, which gives a nice safety buffer for the use of either electric or magnetic fields for neural stimulation, especially when local focusing may reduce the overall illumination power required.

The theoretical efficacy of MW focusing in biological tissue has been demonstrated in this work. For future research, studies at the neuronal circuit level and at the animal behavioral scale are required to better understand the underlying modulation mechanisms to fully utilize this technology. Detailed experimental explorations will be required to understand the excitation mechanisms for precise neural modulation and controls.

## Conclusion

A novel methodology for non-invasive 3D microwave focusing inside biological tissue has been presented. The potential application space of the proposed method is broad and ranges from neural stimulation to tumor ablation. Using an array of magnetic dipoles, it is shown that a desired electric field can be approximated by computing numerical Green's functions and utilizing a regularized least square (RLS) optimization formulation. The method was shown to successfully produce a microwave hotspot in spherical and cylindrical geometries with realistic bioelectrical material properties. Potential improvements to the methodology were discussed and recommended for future studies. The results demonstrate a proof-of-concept for non-invasive and focused stimulation or ablation of biological tissue using engineered microwave signals.

## Author Contributions

**Conceptualization:** Vijay Harid, Hoyoung Kim, Tim Lei.

**Data curation:** Hoyoung Kim.

**Formal analysis:** Vijay Harid.

**Investigation:** Vijay Harid, Hoyoung Kim.

**Methodology:** Vijay Harid, Tim Lei.

**Software:** Vijay Harid, Hoyoung Kim.

**Supervision:** Vijay Harid.

**Visualization:** Ben-Zheng Li.

**Writing – original draft:** Vijay Harid, Hoyoung Kim, Ben-Zheng Li, Tim Lei.

**Writing – review & editing:** Vijay Harid, Hoyoung Kim, Ben-Zheng Li, Tim Lei.

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
