## [Decision Letter · Decision Letter 0]

1 Apr 2022

PONE-D-22-06737A Method for non-destructive microwave focusing for deep brain and tissue stimulationPLOS ONE

Dear Dr. Harid,

Thank you for submitting your manuscript to PLOS ONE. After careful consideration, we feel that it has merit but does not fully meet PLOS ONE’s publication criteria as it currently stands. Therefore, we invite you to submit a revised version of the manuscript that addresses the points raised during the review process.

ACADEMIC EDITOR: In view of the criticism by the reviewers, the manuscript cannot be accepted in its current form. The manuscript may be resubmitted after incorporating all the changes suggested by the reviewers. In a resubmission, the figures must be displayed in good resolution to allow thorough peer-review.

We look forward to receiving your revised manuscript.

Kind regards,

Muhammad Zubair

Academic Editor

PLOS ONE

Journal Requirements:

Additional Editor Comments:

In view of the criticism by the reviewers, the manuscript cannot be accepted in its current form. The manuscript may be resubmitted after incorporating all the changes suggested by the reviewers. In a resubmission, the figures must be displayed in good resolution to allow thorough peer-review.

Reviewers' comments:

Reviewer's Responses to Questions

**Comments to the Author**

1. Is the manuscript technically sound, and do the data support the conclusions?

Reviewer #1: Yes

Reviewer #2: No

2. Has the statistical analysis been performed appropriately and rigorously? 

Reviewer #1: Yes

Reviewer #2: I Don't Know

3. Have the authors made all data underlying the findings in their manuscript fully available?

Reviewer #1: No

Reviewer #2: No

4. Is the manuscript presented in an intelligible fashion and written in standard English?

Reviewer #1: Yes

Reviewer #2: Yes

5. Review Comments to the Author

Reviewer #1: In this manuscript, Harid and colleagues developed a theoretical framework to flexibly use an array of microwave antennas to selectively stimulate body parts with higher spatial resolution than other techniques using lower frequency stimulation. This study is timely, since there is an increased interest in non-invasive stimulation techniques/approaches that can achieve higher spatial resolution.

While I think the manuscript has the potential to be appealing for a large audience, major improvements are required.

Major comments

A - I think the authors should provide more technical details. In particular, the absence of a methods section is unusual. For example, while I do not think that a full derivation of the equations used in this manuscript is necessary, it would help the reader to get a quantitative sense of how they can be derived. Furthermore, some details are actually completely missing. For example, unless I missed that, I did not find how the numerical estimation of the Green functions is computed (also, was any specific software used for that?). Thus, I think that in general this manuscript will benefit from having more technical details in a methods section.

B - The electric fields that are reported in the main figures are extremely weak. A peak amplitude of 0.015 V/m represents a weak field, even compared to the ones induced by transcranial direct current stimulation (tDCS) or transcranial alternating current stimulation (tACS), which are in the order of max 1 V/m (already ~100 times lower than the ones induced in the brain during TMS). In the field of transcranial stimulation there is an ongoing discussion on whether fields of this amplitude are strong enough to induce any effect on neuronal activity at all. Can the intensity of the stimulation reported in this study be increased to achieve higher amplitude electric fields? What are the safety limitations (max intensity that can be used without inducing tissue damage, likely due to heating)? These questions are quite critical to establish the feasibility of MW stimulation in real-case scenarios.

C - I think the discussion should be improved. For instance, what are the benefits of MW stimulation compared, for example, to transcranial ultrasound stimulation (for example Folloni et al., Neuron 2019) or temporal interference (TI) stimulation (for example, Grossman et al., Cell 2017)? It seems to me that all these techniques try to achieve higher spatial resolutions, so a direct comparison would enrich this manuscript. In general a discussion about the tradeoff between focality/intensity/safety (related to point B) is quite necessary in this manuscript.

D - Since a Methods section is missing, so is the “Data Availability Statement” which should describe where the data/code used/generated in this study will be found.

Minor comments

1 - Line 24: “For instance, Deep Brain Stimulation (DBS) requires incision of an extended electrode deep into…”.

Incision -> insertion?

2 - Line 60: “relies on the surgical incision of a MW antenna”.

Incision -> insertion?

3 - Line 63: “For instance, (20) showed that modulated MW signals can also be used to vary firing rates of neurons in mice”.

I think reference 20 is not the correct one.

4 - Line 70: “However, three-dimensional MW focusing within biological tissue has not been adequately investigated in the past due to the complexity of modeling MW signal propagation through the human body”.

The authors should mention exactly why it is complex to model MW signal propagation through the human body.

5 - Line 95: “by the electrical conductivity , dielectric permittivity , conductivity σ,”

Conductivity is mentioned twice.

6 - Line 100: equation 1.

Is \\mu the magnetic permeability? If so, it is not defined in the text.

7 - Line 139: “numerical Green’s functions using FEKO,”

“FEKO” is not defined.

8 - Line 297: “appropriately driving a surrounding the region by an array of antennas”

Something is missing in that sentence. I think the authors meant “appropriately driving an array of antennas surrounding the region”.

9 - Line 326: “For instance, surgeries of deep brain stimulation are currently only reserved for on patients who have developed advanced drug-resistive symptoms due to high invasiveness.”

This sentence is not very clear (and there are some typos). I think that the “invasiveness” (I guess of the electrodes implantation) is not what causes the development of “drug-resistive symptoms”.

10 - Caption Fig.8: “The material is a brain”

I would not call the brain a “material”. I think the authors meant “gray matter” (if the parameters used correspond to gray matter properties)?

Reviewer #2: A proper review cannot be done, the results are so poorly displayed that I cant evaluate the results.

Figures should be improved.

A proper review cannot be done, the results are so poorly displayed that I cant evaluate the results.

Figures should be improved.

6. PLOS authors have the option to publish the peer review history of their article (what does this mean?). If published, this will include your full peer review and any attached files.

Reviewer #1: No

Reviewer #2: No

---

## [Author Response · Author response to Decision Letter 0]

30 May 2022

All responses have been included in the Reviewer Responses document.

To the second reviewer:

The figures should be at a legible quality. If there is an image resolution issue, please contact the editor and we will work with them accordingly. Thank you!

---

## [Decision Letter · Decision Letter 1]

23 Aug 2022

PONE-D-22-06737R1A Method for non-destructive microwave focusing for deep brain and tissue stimulationPLOS ONE

Dear Dr. Harid,

Thank you for submitting your manuscript to PLOS ONE. After careful consideration, we feel that it has merit but does not fully meet PLOS ONE’s publication criteria as it currently stands. Therefore, we invite you to submit a revised version of the manuscript that addresses the points raised during the review process.

Please revise the manuscript in the light of comments mentioned by reviewers; especially, the quality of figures needs improvements. The revised submission addressing all the concerns of reviewers may then be accepted for publication.

We look forward to receiving your revised manuscript.

Kind regards,

Muhammad Zubair

Academic Editor

PLOS ONE

Journal Requirements:

Reviewers' comments:

Reviewer's Responses to Questions

**Comments to the Author**

1. If the authors have adequately addressed your comments raised in a previous round of review and you feel that this manuscript is now acceptable for publication, you may indicate that here to bypass the “Comments to the Author” section, enter your conflict of interest statement in the “Confidential to Editor” section, and submit your "Accept" recommendation.

Reviewer #1: All comments have been addressed

Reviewer #3: (No Response)

2. Is the manuscript technically sound, and do the data support the conclusions?

Reviewer #1: Yes

Reviewer #3: Yes

3. Has the statistical analysis been performed appropriately and rigorously? 

Reviewer #1: Yes

Reviewer #3: N/A

4. Have the authors made all data underlying the findings in their manuscript fully available?

Reviewer #1: Yes

Reviewer #3: Yes

5. Is the manuscript presented in an intelligible fashion and written in standard English?

Reviewer #1: Yes

Reviewer #3: Yes

6. Review Comments to the Author

Reviewer #1: In this revision Harid and colleagues properly addressed all my previous comments. I now understand the methods much better and I am therefore quite pleased with this revised version of the manuscript.

Thank you very much for addressing all my comments.

Just a small detail:

Line 146: “to be calculated using an electromagnetic simulation code.”

I would write software/program instead of “code”.

Reviewer #3: The authors addressed the remarks from reviewer #1. For reviewer #2, the figures are still not well visible in the core of the article but I could download them one by one separately. However, since the background is transparent some axis labels are not visible (especially figs 6, 7, 8 ... h, g, l; d, i, n and; e, j, o). It would require to have the figure separately with white background.

As a brief review, the article is of interesting content for exploring the feasibility of creating MW focal spots inside the human body for either neuro-stimulation or potentially non invasive MW thermal ablation. A few minor revisions need to be adressed:

> fig 6 / 7 / 8 : what does 15n / 10n / 5n mean ?

The goal is to achieve 1V / m at the focal spot however values are much below in the simulation (c, h, m and d, i, n). How do you explain this ?

> as said before, the figures are not easily visible since the background is transparent, it is hard to see the labels when downloading them one by one

> what are the phase shifts necessary to apply for each individual antenna to obtain constructive interference at target ?

> some hardware consideration would be very interesting to discuss. Being able to control the power and the phase shifts of 64 individual antennas at 1GHz seems to be a challenging problem.

> L.377: "the technique requires good probe contact": would not it be the same with the antennas ?" at 1GHz, would the electric field be able to propagate without good contact ? I may be wrong but in thermal ablation, bad tissue contact means high reflected power and low transmission due to permittivity mismatch between air and tissue.

7. PLOS authors have the option to publish the peer review history of their article (what does this mean?). If published, this will include your full peer review and any attached files.

Reviewer #1: No

Reviewer #3: **Yes: **Thomas Bancel

---

## [Author Response · Author response to Decision Letter 1]

15 Oct 2022

Responses have been included in a separate "Response to Reviewers" document.

---

## [Decision Letter · Decision Letter 2]

23 Nov 2022

A Method for non-destructive microwave focusing for deep brain and tissue stimulation

PONE-D-22-06737R2

Dear Dr. Harid,

We’re pleased to inform you that your manuscript has been judged scientifically suitable for publication and will be formally accepted for publication once it meets all outstanding technical requirements.

Kind regards,

Muhammad Zubair

Academic Editor

PLOS ONE

Additional Editor Comments (optional):

Reviewers' comments:

Reviewer's Responses to Questions

**Comments to the Author**

1. If the authors have adequately addressed your comments raised in a previous round of review and you feel that this manuscript is now acceptable for publication, you may indicate that here to bypass the “Comments to the Author” section, enter your conflict of interest statement in the “Confidential to Editor” section, and submit your "Accept" recommendation.

Reviewer #1: All comments have been addressed

Reviewer #3: All comments have been addressed

2. Is the manuscript technically sound, and do the data support the conclusions?

Reviewer #1: (No Response)

Reviewer #3: Yes

3. Has the statistical analysis been performed appropriately and rigorously? 

Reviewer #1: (No Response)

Reviewer #3: N/A

4. Have the authors made all data underlying the findings in their manuscript fully available?

Reviewer #1: (No Response)

Reviewer #3: No

5. Is the manuscript presented in an intelligible fashion and written in standard English?

Reviewer #1: (No Response)

Reviewer #3: Yes

6. Review Comments to the Author

Reviewer #1: (No Response)

Reviewer #3: Thank you for the additionnal information, the hardware consideration was insightful. The article is now acceptable for publication.

7. PLOS authors have the option to publish the peer review history of their article (what does this mean?). If published, this will include your full peer review and any attached files.

Reviewer #1: No

Reviewer #3: **Yes: **Thomas Bancel

---

## [Editor Report · Acceptance letter]

27 Jan 2023

PONE-D-22-06737R2 

A Method for non-destructive microwave focusing for deep brain and tissue stimulation 

Dear Dr. Harid:

I'm pleased to inform you that your manuscript has been deemed suitable for publication in PLOS ONE. Congratulations! Your manuscript is now with our production department. 

Kind regards, 

on behalf of

Dr. Muhammad Zubair 

Academic Editor

PLOS ONE